# Norketamine, the Main Metabolite of Ketamine, Induces Mitochondria-Dependent and ER Stress-Triggered Apoptotic Death in Urothelial Cells via a Ca^2+^-Regulated ERK1/2-Activating Pathway

**DOI:** 10.3390/ijms23094666

**Published:** 2022-04-23

**Authors:** Jhe-Wei Lin, Yi-Chun Lin, Jui-Ming Liu, Shing-Hwa Liu, Kai-Min Fang, Ren-Jun Hsu, Chun-Fa Huang, Kai-Yao Chang, Kuan-I Lee, Kai-Chih Chang, Chin-Chuan Su, Ya-Wen Chen

**Affiliations:** 1Department of Physiology, School of Medicine, College of Medicine, China Medical University, Taichung 404, Taiwan; auakau4567@gmail.com; 2Department of Infectious Diseases, Taoyuan General Hospital, Ministry of Health and Welfare, Taoyuan 330, Taiwan; jean640514@yahoo.com.tw; 3Department of Urology, Taoyuan General Hospital, Ministry of Health and Welfare, Taoyuan 330, Taiwan; mento1218@gmail.com; 4Department of Obstetrics and Gynecology, Tri-Service General Hospital, National Defense Medical Center, Taipei 114, Taiwan; 5Institute of Toxicology, College of Medicine, National Taiwan University, Taipei 100, Taiwan; shinghwaliu@ntu.edu.tw; 6Department of Otolaryngology, Far Eastern Memorial Hospital, New Taipei City 220, Taiwan; u701048@gmail.com; 7Department of Pathology and Graduate Institute of Pathology and Parasitology, Tri-Service General Hospital, Taipei 114, Taiwan; hsurnai@gmail.com; 8School of Chinese Medicine, College of Chinese Medicine, China Medical University, Taichung 404, Taiwan; cfhuang@mail.cmu.edu.tw; 9Department of Nursing, College of Medical and Health Science, Asia University, Taichung 413, Taiwan; 10Department of Emergency, Taichung Tzu Chi Hospital, Buddhist Tzu Chi Medical Foundation, Taichung 427, Taiwan; vltavac@tzuchi.com.tw (K.-Y.C.); leeguanto2002@tzuchi.com.tw (K.-I.L.); 11Center for Digestive Medicine, Department of Internal Medicine, China Medical University Hospital, Taichung 404, Taiwan; d22101@mail.cmuh.org.tw; 12Department of Otorhinolaryngology, Head and Neck Surgery, Changhua Christian Hospital, Changhua County 500, Taiwan

**Keywords:** norketamine, bladder urothelial cell, mitochondrial dysfunction, ER stress, extracellular signal-regulated kinases 1/2, intracellular calcium

## Abstract

Ketamine-associated cystitis is characterized by suburothelial inflammation and urothelial cell death. Norketamine (NK), the main metabolite of ketamine, is abundant in urine following ketamine exposure. NK has been speculated to exert toxic effects in urothelial cells, similarly to ketamine. However, the molecular mechanisms contributing to NK-induced urothelial cytotoxicity are almost unclear. Here, we aimed to investigate the toxic effects of NK and the potential mechanisms underlying NK-induced urothelial cell injury. In this study, NK exposure significantly reduced cell viability and induced apoptosis in human urinary bladder epithelial-derived RT4 cells that NK (0.01–0.5 mM) exhibited greater cytotoxicity than ketamine (0.1–3 mM). Signals of mitochondrial dysfunction, including mitochondrial membrane potential (MMP) loss and cytosolic cytochrome c release, were found to be involved in NK-induced cell apoptosis and death. NK exposure of cells also triggered the expression of endoplasmic reticulum (ER) stress-related proteins including GRP78, CHOP, XBP-1, ATF-4 and -6, caspase-12, PERK, eIF-2α, and IRE-1. Pretreatment with 4-phenylbutyric acid (an ER stress inhibitor) markedly prevented the expression of ER stress-related proteins and apoptotic events in NK-exposed cells. Additionally, NK exposure significantly activated JNK, ERK1/2, and p38 signaling and increased intracellular calcium concentrations ([Ca^2+^]_i_). Pretreatment of cells with both PD98059 (an ERK1/2 inhibitor) and BAPTA/AM (a cell-permeable Ca^2+^ chelator), but not SP600125 (a JNK inhibitor) and SB203580 (a p38 inhibitor), effectively suppressed NK-induced mitochondrial dysfunction, ER stress-related signals, and apoptotic events. The elevation of [Ca^2+^]_i_ in NK-exposed cells could be obviously inhibited by BAPTA/AM, but not PD98059. Taken together, these findings suggest that NK exposure exerts urothelial cytotoxicity via a [Ca^2+^]_i_-regulated ERK1/2 activation, which is involved in downstream mediation of the mitochondria-dependent and ER stress-triggered apoptotic pathway, consequently resulting in urothelial cell death. Our findings suggest that regulating [Ca^2+^]_i_/ERK signaling pathways may be a promising strategy for treatment of NK-induced urothelial cystitis.

## 1. Introduction

Ketamine, a noncompetitive *N*-methyl-D-aspartate (NMDA) receptor antagonist, is usually used as a rapid-onset, short-duration anesthetic and analgesic in clinical surgery [1]. It has been reported that long-term ketamine abuse can cause damage of the upper and lower urinary tract, resulting in the development of ketamine-associated cystitis, which is characterized by urinary frequency/urgency, hematuria and dysuria, urothelial ulceration, inflammatory cell infiltration, and a thickened, contracted, and inflamed bladder [2,3,4]. In histopathological studies, the sloughing/loss of the bladder epithelium (urothelium) was observed in ketamine-associated cystitis patients and mice [5,6]. A growing volume of in vitro studies have shown that exposure to ketamine can induce the urothelial cell cytotoxicity and apoptosis [7,8]. In vivo studies have shown that ketamine promotes oxidative stress-mediated signaling pathways, which are accompanied by mitochondrial dysfunction and endoplasmic reticulum (ER) stress, leading to apoptosis in the bladder [9,10]. These studies suggest that ketamine is directly toxic to the bladder epithelium.

More importantly, norketamine (NK), the main metabolite of ketamine, is generated through ketamine metabolism by cytochrome P450 family enzymes and N-demethylation in the liver [11]. In most cases, NK is eventually hydroxylated to hydroxynorketamine and excreted in the urine following conjugation with glucoronate [12]. However, it is worth noting that a significantly low ratio of urine ketamine/NK (mean 0.743) has been observed under long-term use or abuse ketamine [13]. Therefore, details of the urothelial toxicities relating to the long-term accumulation of NK, as with ketamine, in the urinary bladder, which may cause the cell death and detachment of the urothelium [12,14], needs to be clarified urgently. Several studies have indicated that ketamine exposure can induce urothelial cell cytotoxicity and apoptosis via the oxidative stress/endoplasmic reticulum (ER) stress-mediated pathway [8,9,10,15]. However, to our best knowledge, there are no studies on the toxicological effects and possible mechanisms underlying NK-induced urothelial cell damage.

Mitochondria produce and supply ATP through oxidative phosphorylation processes for the maintenance of cell function and viability [16]. Mitochondria can also play important roles in the regulation of various cellular processes, including the synthesis of key molecules and hormones involved in immune response, and cellular proliferation and apoptosis [17,18,19]. Mitochondrial dysfunction induced by the environmental and toxic insults acting as stimuli has been reported to play a key role in activating apoptosis in mammalian cells, constituting a major risk factor for the development of numerous diseases [18,20]. Some studies have shown that ketamine exposure can induce apoptosis in urothelial cells through triggering cytochrome c release from mitochondria and subsequent activation of caspase-9 and -3 [7,9,21]. However, the role of NK in activating the mitochondria-related apoptotic pathway in urothelial cells remains unclear.

The endoplasmic reticulum (ER) is an organelle widely found in the cells of various eukaryotes that has essential functions in synthesis and correcting folding during the production of secretory proteins [22]. Under adverse environmental challenges, such as pathological states and toxicants exposure, unfolded or misfolded proteins can accumulate in the ER lumen, which perturbs ER homeostasis and triggers the unfolded protein response (UPR), leading to ER stress [23,24]. The UPR is mediated through three independent ER transmembrane receptors: activating transcription factor (ATF)-6, inositol-requiring enzyme 1 (IER1), and protein kinase RNA-like endoplasmic reticulum kinase (PERK) [24,25]. In resting cells, these ER stress receptors are inactivated by binding to the ER chaperone, 78 kDa glucose regulated protein (GRP78). During ER stress, the increase in unfolded/misfolded proteins accumulation causes the dissociation of GRP78 and activation of the three ER stress receptors, triggering the UPR [23,24]. In most cases, the activation of the UPR is a pro-survival response aimed at reducing the backlog of unfolded/misfolded proteins and restoring ER homeostasis. However, if the UPR fails to resolve the ER stress, this otherwise protective signaling switches to eliciting apoptotic responses, resulting in cellular dysfunction and death [22,26]. It has been indicated that the ER stress-initiated apoptosis pathway is linked to toxicant-induced mammalian cell death [27,28,29,30]. Recently, Mansouri et al. [31] observed increased caspase-3 cleavage as well as significant upregulation of CCAAT/enhancer-binding protein (C/EBP) homologous protein (CHOP)/GADD153 mRNA expression in ketamine-treated adult neural stem cells. In vivo and in vitro studies have also shown that ketamine can cause the uroepithelial cell cytotoxicity via induction of apoptosis and the expression of ER stress-related markers [10,32]. However, to our best knowledge, there are no studies that have investigated whether NK can interfere with mitochondrial and/or ER function and cause mitochondria-dependent/ER stress-triggered apoptosis in uroepithelial cells.

Mitogen-activated protein kinases (MAPKs), comprising of the extracellular signal-regulated kinases 1/2 (ERK1/2), c-Jun N-terminal kinase (JNK), p38, and ERK5, are a family of serine/threonine protein kinases that play a crucial role in converting extracellular stimuli into responses affecting a wide range of cellular processes, including cell proliferation, differentiation, growth, and apoptosis [33,34]. Aberrant or inappropriate functions of MAPKs have been widely reported to play important roles in mammalian cell dysfunction, apoptosis, and death and can be induced by various types of stress, such as toxic insults, pathological processes, and intracellular calcium ion ([Ca^2+^]_i_) overload [35,36,37,38,39]. [Ca^2+^]_i_, a ubiquitous and versatile intracellular second messenger, plays important roles in controlling the cellular activities in mammalian cells, involved in multiple signaling cascades, such as those controlling myofilament contraction, hormone/growth factor secretion, metabolism modulation, and gene transcription [40,41]. Excessive and uncompensated increases in [Ca^2+^]_i_ or severe [Ca^2+^]_i_ dysregulation can induce cellular homeostasis loss and disrupt Ca^2+^ signaling, leading to the induction of mitochondria- and ER stress-mediated apoptosis in response to various pathological conditions and chemically induced toxicities [36,42,43,44]. It has been shown that toxic chemicals, such as cadmium and fluorochloridone, can significantly induce [Ca^2+^]_i_ elevation-dependent apoptosis through a MAPK-mediated pathway, leading to cell death [36,38,39]. A growing number of studies have reported that exposure to ketamine induces cell damage and apoptotic death both in vitro and in vivo in urothelial and bladder smooth muscle cells, which were associated with changes in the phosphorylation status of JNK, ERK1/2, and p38 proteins [10,45,46]. Studies by Baker et al. [7] and Zuo et al. [47] also showed that cytosolic [Ca^2+^]_i_ overloads, which induces the deleterious processes leading to apoptosis, are considered a target of the toxic action of ketamine in neuronal and urothelial cells. However, the effects of NK on the activation of MAPKs and [Ca^2+^]_i_ signals in urothelial cells are still unclear.

In this study, our overall aim was to investigate the toxic effects and possible mechanisms of the main ketamine metabolite, NK, on bladder urothelial cells. The roles of MAPKs and [Ca^2+^]_i_ signals and the involvement of mitochondrial dysfunction and the ER stress response in NK-induced urothelial cytotoxicity were examined and clarified.

## 2. Results

### 2.1. Effects of Ketamine and Its Metabolite Norketamine (NK) on Cell Viability and Apoptosis in RT4 Cells

We first investigated the cytotoxic effects of ketamine in human urinary bladder epithelial-derived RT4 cells by MTT assay. Treatment of cells with 0.01–3 mM ketamine for 24 h significantly reduced cell viability in a dose-dependent manner, and the median effective concentration (EC_50_) was approximately 2 mM (52.4 ± 7.4% of control; Figure 1). Next, the caspase-3 activity and caspase cascade activation in ketamine-treated cells were assayed. As shown in Figure 2A, ketamine (1–3 mM) markedly increased caspase-3 activity in a concentration-dependent manner in RT 4 cells following 24 h exposure. Moreover, the treatment of cells with ketamine (2 mM) at various time intervals (8–24 h) resulted in the significant increase in the cleaved forms of caspase-3, -7, -9, and PARP (Figure 2B).

We next examined the cytotoxic effects of NK on RT4 cells. As shown in Figure 1, treating cells with NK for 24 h dramatically decreased cell viability in a concentration-dependent manner (range of 0.01–0.5 mM; 0.01 mM, 92.4 ± 2.8% of control; 0.03 mM, 89.2 ± 2.3% of control; 0.05 mM, 85.7 ± 4.1% of control; 0.1 mM, 74.2 ± 4.4% of control; 0.2 mM, 55.3 ± 4.1% of control; 0.3 mM, 31.3 ± 4.6% of control; 0.5 mM, 9.8 ± 1.9% of control; *p* < 0.05). The NK EC_50_ for cell viability inhibition was approximately 0.2 mM. Moreover, treatment of RT4 cells with NK (0.1–0.3 mM) for 24 h also significantly increased caspase-3 activity (at 0.2 mM, 194.1 ± 24.2% of control), which could effectively be abolished by the addition of Z-VAD-FMK (3 μM; the pan-caspase inhibitor) (Figure 3A). Based on these results, the concentration of 0.2 mM NK was chosen for use in subsequent experiments.

We next tested the effects of NK on the activation of caspase cascade proteases, which are one of the most widely recognized apoptotic markers. As shown in Figure 3B, Western blot analysis displayed an obvious increase in the protein expression of cleaved forms of caspase-3, -7, and -9 and PARP in NK (0.2 mM)-treated RT4 cells following 8–24 h exposure. Pretreatment of cells with Z-VAD-FMK (3 μM) for 1 h prior to NK exposure for 24 h significantly reduced caspase-3, -7, and PARP activation (Figure 3C). These results indicate that treatment of RT4 cells with NK is capable of inducing apoptosis. Moreover, NK has a more potent cytotoxic effect than ketamine on RT4 cells.

### 2.2. Norketamine (NK)-Induced Apoptosis Is Mediated by a Mitochondria-Dependent Pathway in RT4 Cells

To investigate whether NK-induced cell apoptosis was mediated by a mitochondria-dependent pathway, the MMP and cytochrome *c* release were analyzed. As shown in Figure 4A,B, treatment of cells with NK (0.2 mM) for 6 h led to statistically significant reduction in MMP (87.3 ± 3.9% of control), while there was a more marked reduction following 24 h treatment (64.1 ± 6.3% of control). Cytochrome c release from mitochondria into the cytosolic fraction was also slightly (for 6 h) and markedly (for 24 h) increased in NK (0.2 mM)-treated RT4 cells (Figure 4C). These results indicate that the mitochondria-dependent apoptotic pathway may play an important role in NK-induced RT4 cell apoptosis.

### 2.3. Norketamine (NK) Induces the ER Stress Response in RT4 Cells

To determine whether ER stress signals were involved in NK-induced cell apoptosis, the protein expression of ER stress-related markers was analyzed. As shown in Figure 5A, treatment of cells with NK (0.2 mM) for 8–24 h obviously triggered the protein expression of ER stress-related molecules, including GRP78, CHOP, the spliced form of XBP-1 (XBP-1s), ATF-4, cleaved ATF-6, and caspase-12 (the downstream molecule of ER stress); but NK did not affect the expression of GRP 94 protein. In addition, the levels of phosphorylated PERK, eIF2α, and IRE1 (the major arms of ER stress) were significantly increased in RT4 cells treated with NK for 2–8 h (Figure 5B).

To further confirm the relationship between the activation of ER stress and cell apoptosis, an ER stress inhibitor 4-phenylbutyric acid (4-PBA) was used. As shown in Figure 5C,D, pretreatment of cells with 4-PBA (2 mM) for 1 h prior to NK exposure for 24 h effectively and significantly inhibited both upregulation of ER stress-related molecules (including GRP 78, CHOP, XBP-1s, ATF-4, cleaved ATF-6, and cleaved caspase-12) and apoptotic events (including cleaved caspase-3 and -7, and PARP proteins and caspase-3 activity). These results imply that NK is capable of inducing ER stress responses, with downstream regulation of RT4 cell apoptosis.

### 2.4. ERK1/2 Signaling Pathway Plays an Important Role in Norketamine (NK)-Induced RT4 Cell Apoptosis

The MAPK signaling pathway is well-known to be regulated by various chemicals that induce cytotoxicity and apoptosis in mammalian cells. We next investigated whether the activation of MAPKs signals was involved in NK-induced RT4 cell apoptosis. As shown in Figure 6, the levels of phosphorylated JNK1/2, ERK1/2, and p38 proteins were effectively increased in RT4 cells treated with NK (0.2 mM) for 30–90 min. Pretreatment of cells with a pharmacological inhibitor of ERK1/2 (PD98058; 20 μM), but not with inhibitors of either JNK (SP600125) or p38 (SB203580), for 1 h prior to NK exposure effectively attenuated this reduction in cell viability (Figure 7A) and the induction of apoptotic events (Figure 7B,C) and increase in ER stress-related molecules (Figure 7C), mitochondrial dysfunction events (cytosolic cytochrome *c* release, Figure 7D; MMP loss, Figure 7E), and phosphorylation of ERK1/2 (Figure 7F). These results suggest that ERK1/2 signal activation-mediated mitochondria-dependent and ER stress-regulated apoptosis is involved in NK-induced RT4 cell death.

### 2.5. The Role of [Ca^2+^]_i_ Signaling in Norketamine (NK)-Induced RT4 Cell Apoptosis

It has been reported that [Ca^2+^]_i_ signaling is involved in toxic insults-induced apoptosis in various types of cells [35,36,48]. To ascertain whether [Ca^2+^]_i_ signal was involved in NK-induced RT4 cell apoptosis, changes in [Ca^2+^]_i_ were analyzed using flow cytometry. As shown in Figure 8A, treatment of RT4 cells with NK (0.2 mM) for 0.5–2 h triggered a time-dependent increase in the levels of [Ca^2+^]_i_. Pretreatment of cells with the cell-permeable [Ca^2+^]_i_ chelator BAPTA/AM (5 μM) for 1 h prior to NK exposure effectively and significantly prevented [Ca^2+^]_i_ elevation (Figure 8A), the induction of apoptotic events (Figure 8B,C) and increase in ER stress-related molecules (Figure 8C), and cytosolic cytochrome c release (Figure 8D) in NK-treated RT4 cells.

Furthermore, since both ERK1/2 and [Ca^2+^]_i_ signals were found to be involved in the NK-induced RT4 cell apoptosis, we next investigated whether there was the relationship between ERK1/2 and [Ca^2+^]_i_ signaling. As shown in Figure 8E, pretreatment of cells with BAPTA/AM (5 μM) for 1 h prior to NK exposure significantly abrogated ERK1/2 protein phosphorylation. However, the pretreatment of RT4 cells with PD98059 (20 μM) could not prevent the NK-induced [Ca^2+^]_i_ elevation (Figure 8A). These results suggest that [Ca^2+^]_i_ plays a crucial role in ERK1/2 signaling regulation of both mitochondria-dependent and ER stress-triggered apoptosis, contributing to NK-induced TR4 cell death.

## 3. Discussion

In this study, we first observed that NK induced cytotoxicity in urothelial cells with an EC_50_ of approximately 0.2 mM. We further demonstrated that treatment of NK (0.1–0.3 mM) for 24 h could induce urothelial cell apoptosis. Ketamine, a NMDA receptor antagonist, has been widely used as an anesthetic drug and is being increasingly abused as a recreational drug due to its hallucinatory effects [49,50]. Long-term ketamine abuse has been reported to cause lower urinary tract toxicity (termed ketamine cystitis), which displays pathological features of a contracted bladder with urothelium denudation [51]. It has been indicated that ketamine and its metabolites in the urine of the bladder directly contacts the urothelium, which has been shown to result in cell body shrinkage, chromatin condensation, and layer denudation, leading to urothelial cell apoptosis and focal full-thickness urothelial loss [5,6,15]. Furthermore, it has been determined that ketamine is metabolized in the liver by cytochrome P450 enzymes (CYP3A4, CYP2B6, and CYP2CP) and transformed to NK via the N-demethylation pathway [52]. Chang et al. [14] and Shahani et al. [4] studied patients of ketamine abuse and observed the apoptosis and detachment of bladder urothelium, bladder inflammation, and bladder mucosa damage, which led to ulcerative cystitis. High concentrations of both ketamine and NK in the urine were also detected. The findings of Moore et al. [13] analyzing urine samples of patients following illegal consumption of ketamine showed that there was a low ratio of urine ketamine/NK, ranging from 0.132 to 0.857. In animal models, where ketamine exposure caused ulcerated urothelium and bladder damage, it was also revealed that the levels of NK were higher concentration than those of ketamine in rat urine (ketamine vs. NK for 14 and 28 days, i.p. injection: 1406 ± 241.0 to 1668.0 ± 225.6 ng/mL vs. 30016.0 ± 1161.4 to 33720.2 ± 1262.7 ng/mL; data from Liu et al., [9]) [9,52]. Juan et al. [52] showed that cell viability is reduced after the exposure of primary urothelial cells to 1 mM (about 237,725 ng/mL) ketamine for 24 h, similarly to exposure to urine samples obtained from ketamine-treated rats, which contained 1.4 ng/mL ketamine and 38 ng/mL NK after 1000 dilution. These previous findings reveal that the toxic effects of ketamine on the bladder urothelium are due not only to ketamine but also its main metabolite NK. Juan et al. [53] also observed that treatment of rat primary urothelial cells with a urine sample from ketamine-treated rats, which contains ketamine and NK, can induce cytotoxicity and stimulate the expression of NF-κB-p65 and COX-2 proteins, which may be involved in ketamine-induced ulcerative cystitis in rats. In this study, we further demonstrate that exposure of urothelial cells to NK significantly induced cytotoxicity in a concentration-dependent manner (ranging from 0.01 to 0.5 mM), and that NK exhibits greater cytotoxicity than its parent compound ketamine.

There are many environmental toxicants reported to cause mitochondrial dysfunction, which is accompanied by activation of the mitochondria-mediated caspase pathway (a major apoptosis pathway) characterized by a reduction in mitochondrial membrane potential (ΔΨ_M_) and mitochondrial outer membrane permeabilization (MOMP) with subsequent release of cytochrome c into the cytoplasm, which triggers caspase activation [54,55]. Exposure of pancreatic β-cells to toxic chemicals, such as misfolded human islet amyloid polypeptide (hIAPP), molybdenum, and high-glucose, induces cytotoxicity and cell death, which are mechanistically accompanied by activation of the pathways leading to mitochondrial dysfunction and apoptosis [56,57,58]. Pretreatment with antioxidant N-acetylcysteine can effectively prevent As^3+^- and Cr^6+^-induced mitochondrial toxicities (MMP depolarization and cytosolic cytochrome c release) and subsequent apoptotic responses in neuronal cells [59,60]. A growing number of studies indicates that exposure to ketamine of various mammalian cells, such as hepatocytes, neuronal cells, and urothelial cells, can result in mitochondrial membrane depolarization and an increase in MOMP, allowing the release of cytochrome c into the cytosol, which results in apoptosis [7,9,61,62]. In the present study, we demonstrate the role of mitochondrial dysfunction-mediated apoptosis in NK-induced urothelial cell death.

Chronic/overwhelming ER stress is a major factor affecting the initiation of apoptosis, which can lead to cell death, organ dysfunction, and development of numerous pathological diseases [22,63]. In addition to caspase-12, other ER stress markers, including GRP78 and 94, ATF-4, XBP-1s, and ER-associated apoptosis protein CHOP, have been reported to be upregulated in mammalian cells exposed to toxicants. Typically, in vitro exposure to toxic chemicals, such as MeHg and 4-methyl-2,4-bis(4-hydroxyphenyl)pent-1-ene, in experiments results significantly induced expression of ER stress-related markers in neuronal cells and pancreatic β-cells, which contributes to apoptotic cell death [27,29,64]. Recently, several studies have indicated that ketamine reduces cell viability and elevated apoptosis in urothelial cells and neural stem cells via the activation of ER stress-related markers, including IRE1, GRP78, and CHOP [10,31,32]. Several lines of studies using experimental animal models coupled with results from patients with abuse or illegal consumption of ketamine showed that ketamine injection could induce expression ER stress-related markers, eliciting bladder apoptosis as well as high levels of NK in the urine, suggesting that the toxic effects of ketamine on urinary bladder and urothelial cells were due not only to ketamine but also its main metabolite NK [9,10,13,15]. However, no studies have investigated the effects of NK on ER stress underlying urothelial cell death. In this study, we found that NK was capable of causing the mitochondrial dysfunction as indicated by a loss of MMP and an increase in the release of cytosolic cytochrome *c*, resulting in urothelial cell apoptosis. Furthermore, exposure of urothelial cells to NK obviously increased the expression of ER stress-related molecules. Pretreatment of cells with 4-PBA could effectively attenuate NK-induced ER stress and apoptosis. These results suggest that NK exposure can cause urothelial cell death via both mitochondria-dependent and ER stress-mediated apoptotic pathways.

In this study, our results reveal the significant activation of ERK1/2 protein by NK in urothelial cells. ERK1/2, belong to the MAPK family, plays an important role in regulating a broad range of cellular activities, including cell proliferation, differentiation, transcription, and survival [65,66]. By contrast, ERK1/2 signal induced by exogenous/endogenous stimuli has been demonstrated to play a role in controlling the mitochondrial dysfunction and intrinsic apoptotic pathway, leading to cell death and the development of various diseases [59,64,67,68]. In rats, ketamine exposure has been observed to significantly induce rewarding behavior via conditioned place preference, which is accompanied by marked increases in phosphorylated ERK1/2 protein in the hippocampus and caudate putamen [69]. A study by Chen et al. [45] demonstrated a significant increase in phosphorylated ERK1/2 protein as well as mitochondria damage and apoptosis in the urinary bladder of ketamine-treated rats. Nevertheless, no studies have investigated the role of ERK1/2 in NK-induced urothelial cell death. Herein, our results showed that treatment of RT4 cells with NK resulted in significantly increased phosphorylation of JNK, ERK1/2, and p38 proteins. Pretreatment of cells with the pharmacological ERK1/2 inhibitor PD98059, but not the JNK inhibitor (SP600125) or p38 inhibitor (SB203580), effectively prevented the reduction in cell viability, apoptotic events, mitochondrial dysfunction, expression of ER stress-related molecules, and ERK1/2 activation in NK-treated RT4 cells. These results imply that ERK1/2 activation-mediated mitochondria-dependent and ER stress-triggered apoptotic pathways play an important role in NK-induced urothelial cell death.

ER stress response and mitochondrial dysfunction induced by abnormally increased [Ca^2+^]_i_ have been identified to be involved in apoptotic routes [42,44]. It has been indicated that toxic chemicals, such as methylmercury (MeHg), cadmium (Cd), butylated hydroxyanisole (BHA), and bisphenol AF (BPAF), can induce apoptosis through [Ca^2+^]_i_ overload and mitochondrial dysfunction/ER stress induction in various cell types [35,36,70,71]. Previous studies have also revealed that a significant elevation in [Ca^2+^]_i_ resulting from ketamine exposure is associated with mitochondrial dysfunction and apoptotic cell death in neuronal and urothelial cells [7,72]. However, to the best of our knowledge, no studies have elucidated the important regulators of [Ca^2+^]_i_ in NK-induced urothelial cell death and the sequential relationship between [Ca^2+^]_i_ and ERK1/2 signaling that leads to urothelial cell apoptosis. In the present study, our results showed that NK significantly elevated [Ca^2+^]_i_ levels in RT4 cells. Pretreatment of cells with calcium chelator BAPTA/AM effectively abrogated NK-induced apoptotic events, the expression of ER stress-related molecules, mitochondrial dysfunction, and ERK1/2 protein phosphorylation. Moreover, the elevation of intracellular [Ca^2+^]_i_ in NK-exposed RT4 cells was significantly abolished by BAPTA/AM but not ERK1/2 inhibitor PD98059. These findings suggest that [Ca^2+^]_i_ plays a role as a key regulator of NK-induced cytotoxicity, which causes ERK1/2 activation downstream-mediated mitochondria dysfunction and ER stress activation, contributing to the induction of apoptosis, and therefore urothelial cell death.

Oxidative stress has been demonstrated to play a critical role in several pathological conditions of the urinary bladder [73]. Several recent in vitro and in vivo studies have reported that ketamine abuse can induce reactive oxygen species (ROS) production, leading to uroepithelial cell apoptosis and death [8,9,51,74]. Furthermore, it is well known that the nuclear factor erythroid 2-related factor 2 (Nrf2), a member of a family of basic leucine transcription factors participating in the regulation of antioxidant response element (ARE)-mediated gene expression, is part of one of the most important protective mechanisms/antioxidant defense systems against oxidative stress in mammalian cells [75]. Nrf2 activates a series of enzymes with antioxidant and detoxifying activity that play a key role in the protection of cells against various environmental stresses, such as electrophiles, reactive oxygen species, DNA damage, and apoptosis [76]. Under normal conditions, Nrf2 is present in the cytoplasm and complexes with Kelch-like ECH-associated protein 1 (Keap1). In response to oxidative stress, Nrf2 is dissociated from Keap1 (the conformational change and the release) and then translocates into the nucleus [77]. It has been shown that an increase in Nrf2 levels can protect uroepithelial cells from chemical-induced cytotoxicity and apoptosis [78,79]. Nrf2 knockout substantially increases the susceptibility to a broad range of chemical toxicity and disease conditions associated with oxidative pathology [80,81]. A study by Sun et al. [82] has shown the positive correlation between the loss of Nrf2 and the exacerbation of ER stress-induced apoptosis in the brain tissue of Nrf2 knockout mice with traumatic brain injury. Recently, Liu et al. [9] and Cui et al. [32] reported that ketamine exposure-induced uroepithelial cell apoptosis involves oxidative stress and ER stress responses. Our results from this study also found that NK, similarly to ketamine [9,32], could significantly increase ROS generation (Appendix A) and the expression of ER stress-related molecules (Figure 5). Both ketamine and norketamine treatment also led to a marked decrease in Nrf2 protein levels (Appendix A), suggesting that Nrf2 might play a key role in ketamine- and NK-induced uroepithelial cell apoptosis. However, the relationship between oxidative stress, Nrf2, and ER stress underlying the ketamine- and NK-induced uroepithelial cell injury are mostly unclear. Thus, further experiments are required to investigate this important issue in the future.

## 4. Materials and Methods

### 4.1. Materials

Unless otherwise specified, all chemicals and laboratory plastic wares were purchased from Sigma-Aldrich (St. Louis, MO, USA) and Falcon Labware (Becton, Dickinson and Company, Franklin Lakes, NJ, USA), respectively. McCoy’s 5A Medium, fetal bovine serum (FBS), and antibiotics were purchased from Gibco/Invitrogen (Thermo Fisher Scientific Inc., Waltham, MA, USA). Mouse- or rabbit- monoclonal antibodies specific for cleaved caspase-3 (Cat. No.: #9661), cleaved caspase-7 (Cat. No.: #9491), cleaved caspase-9 (Cat. No.: #7237), PARP (Cat. No.: #9542) cytochrome *c* (Cat. No.: #11940), GRP 78 (Cat. No.: #3183), GRP 94 (Cat. No.: #2104), CHOP (Cat. No.: #2895), XBP-1 (Cat. No.: #40435), ATF-4 (Cat. No.: #11815), PERK (Cat. No.: #3192), phosphorylated (p)-eIF-2α (Cat. No.: #5199), eIF-2α (Cat. No.: #9722), p-JNK (Cat. No.: #9251), p-ERK1/2 (Cat. No.: #4377), p-p38 (Cat. No.: #9216), JNK-1 (Cat. No.: #3708), JNK-2 (Cat. No.: #4672), ERK1/2 (Cat. No.: #9102), p38 (Cat. No.: #8690), and β-actin (Cat. No.: #8457), and secondary antibodies (horseradish peroxidase (HRP)-conjugated anti-mouse IgG (Cat. No.: #7076) or anti-rabbit IgG (Cat. No.: #7074)) were purchased from Cell Signaling Technology (Cell Signaling Technology, Danvers, MA, USA); an antibody specific for p-PERK (Cat. No.: sc-32577) was purchased from Santa Cruz Biotechnology (Santa Cruz Biotechnology, Santa Cruz, CA, USA); an antibody specific for caspase-12 (Cat. No.: ab62484) was purchased from Abcam (Abcam plc, Cambridge, UK); an antibody specific for ATF-6 (Cat. No.: NBP1-40256) was purchased from Novus Biologicals (Novus Biologicals LLC, Littleton, CO, USA); and an antibody specific for p-IRE-1 (Cat. No.: AP0878) was purchased from Abclonal (Woburn, MA, USA).

### 4.2. Cell Culture

RT4 cells, a human urinary bladder epithelial-derived cell line, were purchased from American Type Culture Collection (ATCC; HTB-2) and cultured in a humidified chamber with a 5% CO_2–_95% air mixture at 37 °C and maintained in McCoy’s 5A medium supplemented with 10% fetal bovine serum (FBS) and antibiotics (100 U/mL of penicillin and 100 μg/mL of streptomycin).

### 4.3. Cell Viability Assay

Cells were washed with fresh medium and cultured in 96-well plates (2 × 10^4^ cells/well) and then stimulated with ketamine (0.01–3 mM) or NK (0.01–0.5 mM) for 24 h. After incubation, medium was aspirated and cells were incubated with fresh medium containing 0.2 mg/mL 3-(4,5-dimethyl thiazol-2-yl-)-2,5-diphenyl tetrazolium bromide (MTT). After 4 h, the medium was removed and the blue formazan crystals were dissolved in 100 μL of dimethyl sulfoxide (DMSO). Absorbance at 570 nm was measured using a Bio-Tek uQuant Microplate Reader (MTX Lab Systems, Winooski, VT, USA).

### 4.4. Determination of Caspase-3 Activity

Caspase-3 activity was assessed using a Caspase-3 Activity Assay Kit (Cell Signaling Technology, Inc.). RT4 cells were seeded at 2 × 10^5^ cells/well in a 24-well plate and treated with ketamine (1–3 mM) or NK (0.1–0.3 mM) at 37 °C in the absence or presence of Z-VAD-FMK (3 μM), 4-phenylbutyric acid (4-PBA; 2 mM), SP600125 (20 μM), PD98059 (20 μM), SB203580 (20 μM), and BAPTA/AM (5 μM). At the end of treatment (for 24 h), the cell lysates were incubated at 37 °C with 10 μM Ac-DEVD-AMC, a caspase-3/CPP32 substrate, for 1 h. The fluorescence of the cleaved substrate was measured using a spectrofluorometer (Gemini XPS Microplate Reader, Molecular Devices, San Jose, CA, USA) at an excitation wavelength at 380 nm and an emission wavelength at 460 nm.

### 4.5. Detection of Mitochondrial Membrane Potential (MMP)

MMP was analyzed using the fluorescent probe DiOC_6_, which is a positively charged mitochondria-specific fluorophore. Briefly, RT4 cells were seeded at 2 × 10^5^ cells/well in a 24-well plate and incubated with NK (0.2 mM) for 6 or 24 h in the absence or present (1 h pre-treatment) of PD98059 (20 μM) or BAPTA/AM (5 μM). At the end of treatment, cells were incubated with medium containing 100 nM 3,3′-dihexyloxacarbocyanine iodide (DiOC_6_) for 30 min at 37 °C. After incubation with the dye, cells were harvested and washed twice with phosphate buffered saline (PBS), and then re-suspended in ice-cold PBS. MMP was analyzed using a flow cytometer (FACScalibur, Becton, Dickinson and Company, Franklin Lakes, NJ, USA) and CellQuest software version 5.1 (Becton, Dickinson and Company, Franklin Lakes, NJ, USA).

### 4.6. Western Blot Analysis

RT4 cells were seeded at 1 × 10^6^ cells/well in a 6-well culture plate and treated with ketamine (2 mM) or NK (0.2 mM) in the absence or presence (1 h pre-treatment) of 4-phenylbutyric acid (4-PBA; 2 mM), PD98059 (20 μM), or BAPTA/AM (5 μM). At the end of various treatment durations, the levels of protein expression were analyzed by Western blot analysis as previously described [27,36]. In brief, equal amounts of proteins (50 μg per lane) were subjected to electrophoresis on 10% (*W*/*V*) SDS-polyacrylamide gels and transferred to polyvinylidene difluoride (PVDF) membranes. The membranes were blocked for 1 h in PBST (PBS, 0.05% Tween-20) containing 5% nonfat dry milk. After blocking, the membranes were incubated with mouse- or rabbit-monoclonal antibodies specific against cleaved caspase-3, cleaved caspase-7, cleaved caspase-9, cleaved caspase-12, cleaved PARP, GRP 78, GRP 94, CHOP, XBP-1,ATF-4 and -6, p-PERK, p-eIF2α, p-IRE-1, p-JNK, p-ERK1/2, p-p38, PERK, eIF2α, JNK-1, JNK-2, ERK1/2, p38, and β-actin in 0.1% PBST (1:1000) for 12–16 h at 4 °C. After three additional washes in 0.1% PBST (15 min each), the respective HRP-conjugated secondary antibodies were applied (1:2500 in 0.1% PBST) for 1 h at 4 °C. The antibody-reactive bands were detected using enhanced chemiluminescence reagents (Pierce^TM^, Thermo Fisher Scientific Inc., USA) and analyzed using a luminescent image analyzer (ImageQuant™ LAS-4000) (GE Healthcare Bio-Sciences Corp., Piscataway, NJ, USA). For cytosol cytochrome c expression, the detection was performed as previously described in Fu et al. [59]. In brief, at the end of treatments, cells were detached, washed twice with PBS, and then homogenized in extract buffer (0.4 M mannitol, 25 mM MOPS (pH 7.8), 1 mM EGTA, 8 mM cysteine, and 0.1% (*w*/*v*) bovine serum albumin) using a pestle and mortar. The cell debris was removed via centrifugation at 6000× *g* for 2 min. The supernatant was recentrifuged at 12,000× *g* for 15 min. The supernatant (cytosolic fraction) was detected cytochrome c expression by Western blot analysis.

### 4.7. Measurement of Intracellular Calcium Levels

Intracellular calcium levels were monitored by Fluo-3/AM (Sigma-Aldrich, St. Louis, MO, USA), which is a Ca^2+^-sensitive fluorescent indicator. In brief, RT4 cells were seeded (2 × 10^5^ cells/well) in 24-well culture plates and treated with NK (0.2 mM) in the absence or presence of 5 μM BAPTA/AM or 20 μM PD98059 (prior to the treatment with norketamine). At the end of the treatment period (for 0.5–2 h), the medium was replaced with fresh McCoy’s 5A supplemented with Fluo-3/AM (5 μM), followed by an additional 30 min incubation in the dark. Cells were washed twice with PBS to remove the intracellular AM esters before being analyzed. Fluorescence intensities were detected by flow cytometry (FACScalibur, Becton, Dickinson and Company, Franklin Lakes, NJ, USA) using CellQuest software version 5.1 (Becton, Dickinson and Company, Franklin Lakes, NJ, USA).

### 4.8. Statistical Analysis

Data are presented as the mean ± standard deviation (SD) of at least four independent experiments. All data analyses were performed using SPSS software version 12.0 (SPSS, Inc., Chicago, IL, USA). For each experimental condition, significant differences were assessed by one-way analysis of variance (ANOVA) followed by Tukey’s post hoc test; *p* value < 0.05 was considered to indicate significance.

## 5. Conclusions

In our study, we elucidated that NK can cause urothelial cytotoxicity through the mitochondria-dependent and ER stress-triggered apoptotic pathways. The axis of [Ca^2+^]_i_-activated ERK1/2 signal pathway is identified as a critical mechanism underlying NK-induced urothelial cell apoptosis. These findings not only provide useful evidence supporting the role of the main ketamine metabolite NK as a key substance in ketamine abuse-induced urothelial cytotoxicity, but also underline that the regulation of [Ca^2+^]_i_/ERK signaling pathways may be a promising intervention with potential application for the development of targeted drugs in the treatment of NK-induced urothelial cystitis.

## Figures and Tables

**Figure 1 ijms-23-04666-f001:**
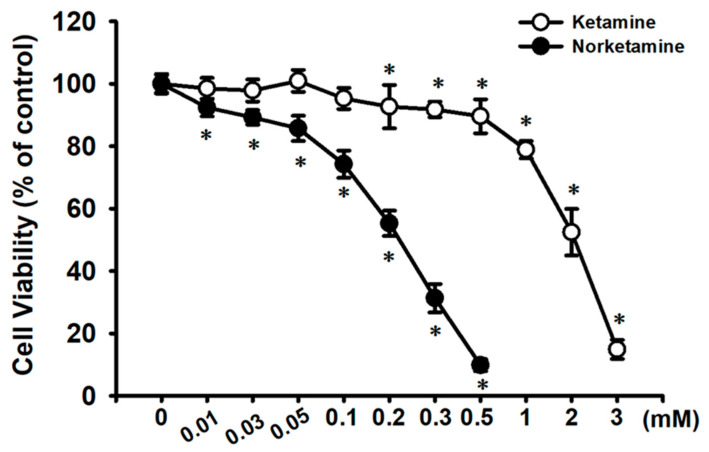
Cytotoxic effect of ketamine and norketamine (NK) on RT4 cells. Cells were treated with ketamine (0.01–3 mM) or NK (0.01–0.5 mM) for 24 h, and cell viability was determined using MTT assay. Data are presented as the means ± SD of four independent experiments assayed in triplicate. * *p* < 0.05 compared to vehicle control.

**Figure 2 ijms-23-04666-f002:**
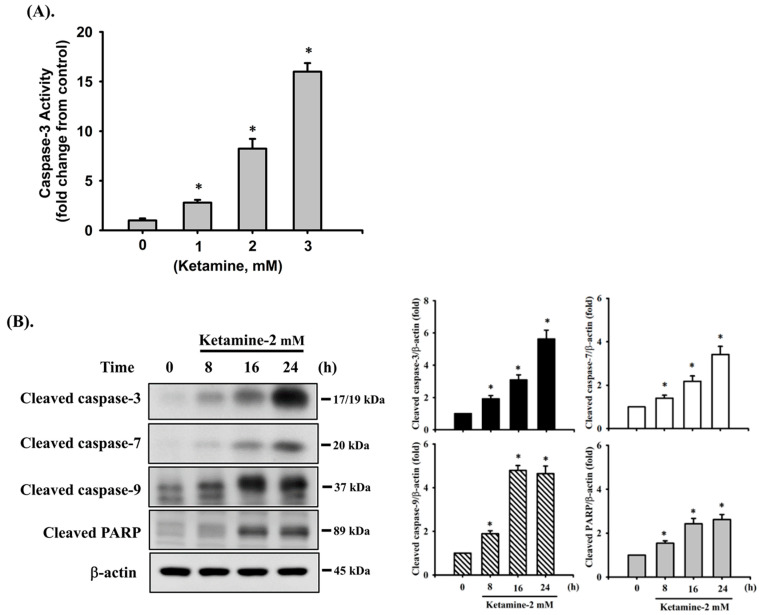
Ketamine induces apoptosis in RT4 cells. Cells were treated with ketamine (2 mM) for various time intervals. (**A**) Caspase-3 activity was detected using the Caspase-3 Activity Assay Kit. Data are presented as the means ± SD of four independent experiments assayed in triplicate. * *p* < 0.05 as compared to vehicle control. (**B**) The levels of protein expression for the precursor form of caspase-3, -7, and -9, and PARP were examined using Western blot analysis. Results shown on a representative image, and quantification was determined by densitometric analysis. Each bar presented is the mean ± SD of three independent experiments. * *p* < 0.05 compared to the vehicle control.

**Figure 3 ijms-23-04666-f003:**
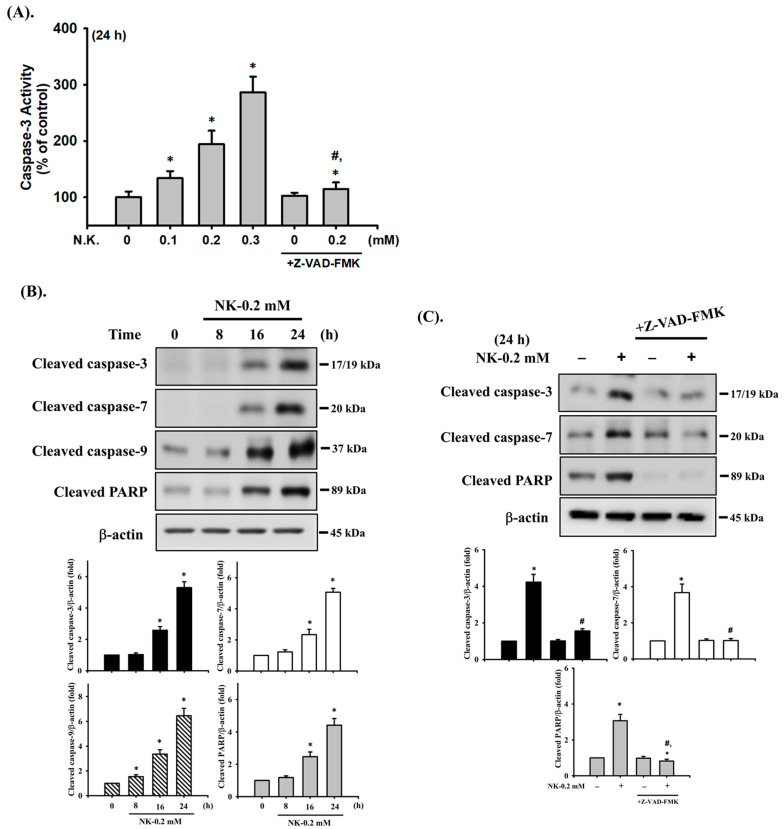
Norketamine (NK) induces apoptosis in RT4 cells. Cells were treated with NK (0.1–0.3 mM) for 8–24 h in the absence or presence of Z-VAD-FMK (3 μM; the pan-caspase inhibitor) for 1 h prior to the addition of NK. (**A**) Caspase-3 activity was detected using the caspase-3 Activity Assay Kit, and data are presented as the means ± SD of four independent experiments assayed in triplicate. (**B**,**C**) The levels of protein expression for the cleaved form of caspase-3, -7, -9, and PARP were examined using Western blot analysis, and results shown on representative images, and quantification was determined by densitometric analysis. Each bar presented is the mean ± SD of three independent experiments. * *p* < 0.05 compared to the vehicle control. # *p* < 0.05 compared to NK treatment alone.

**Figure 4 ijms-23-04666-f004:**
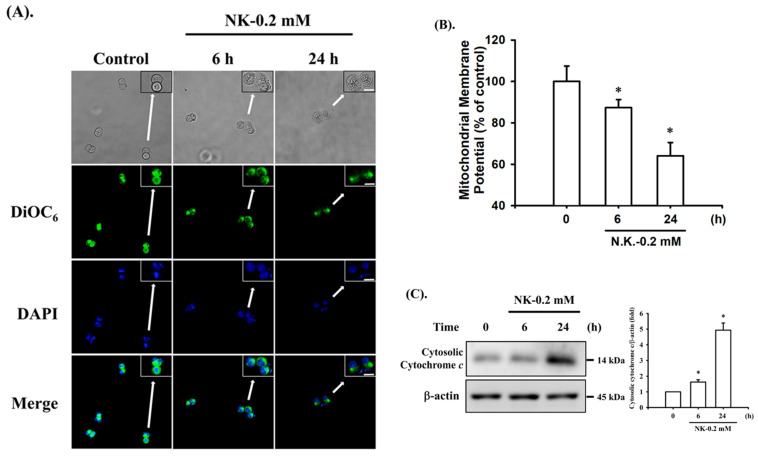
Norketamine (NK) induces mitochondrial dysfunction in RT4 cells. Cells were treated with or without NK (0.2 mM) for 6 and 24 h. (**A**) DiOC_6_ fluorescent probe was used to detect changes in MMP. The images were captured by fluorescence microscopy at a magnification of 400×. Scale bar: 20 μm. (**B**) Loss of MMP was detected and quantified using flow cytometry. Data are presented as the means ± SD of four independent experiments assayed in triplicate. (**C**) The release of cytochrome c from the mitochondria into the cytosolic fraction was analyzed using Western blot analysis. Results shown on a representative image, and quantification was determined by densitometric analysis. Each bar presented is the mean ± SD of three independent experiments. * *p* < 0.05 compared to vehicle control.

**Figure 5 ijms-23-04666-f005:**
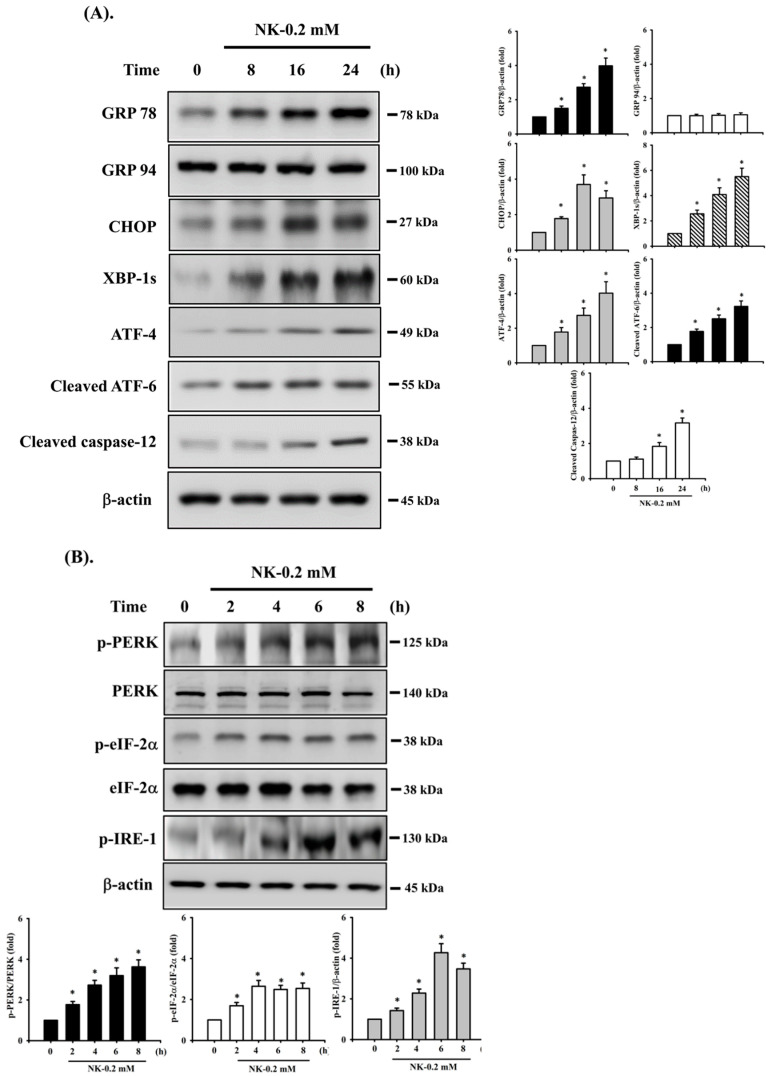
Involvement of ER stress-regulated signaling in norketamine (NK)-induced RT4 cell apoptosis. Cells were treated with NK (0.2 mM) for different time intervals (2–24 h), and the levels of protein expression for (**A**) GRP78, GRP94, CHOP, XBP-1s, ATF-4, cleaved ATF-6, and caspase-12, and (**B**) the phosphorylated or total eIF2α, PERK, and IRE-1 were examined using Western blot analysis. Additionally, RT4 cells were treated with NK (0.2 mM) for 24 h in the absence or presence of an ER stress inhibitor 4-phenylbutyric acid (4-PBA; 2 mM; for 1 h prior to treated with NK), and (**C**) the levels of protein expression for GRP78, CHOP, XBP-1s, ATF-4, cleaved ATF-6, Caspase-3, -7, -12, and PARP were examined using Western blot analysis; and (**D**) caspase-3 activity was detected using the caspase-3 Activity Assay Kit. Results shown in A, B, and C on representative images, and quantification was determined by densitometric analysis. Each bar presented is the mean ± SD of three independent experiments. Data in D are presented as the means ± SD of four independent experiments assayed in triplicate. * *p* < 0.05 compared to vehicle control. # *p* < 0.05 compared to NK treatment alone.

**Figure 6 ijms-23-04666-f006:**
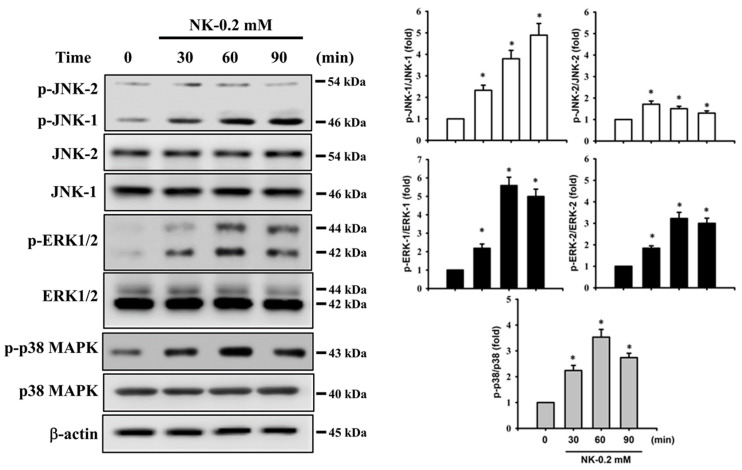
Effects of norketamine (NK) on the phosphorylation of MAPK signals in RT4 cells. Cells were treated with NK (0.2 mM) for 30–90 min, and the protein phosphorylation levels of JNK, ERK1/2, and p38 were examined using Western blot analysis. Results shown on representative images, and quantification was determined by densitometric analysis. Each bar presented is the mean ± SD of three independent experiments. * *p* < 0.05 compared to vehicle control.

**Figure 7 ijms-23-04666-f007:**
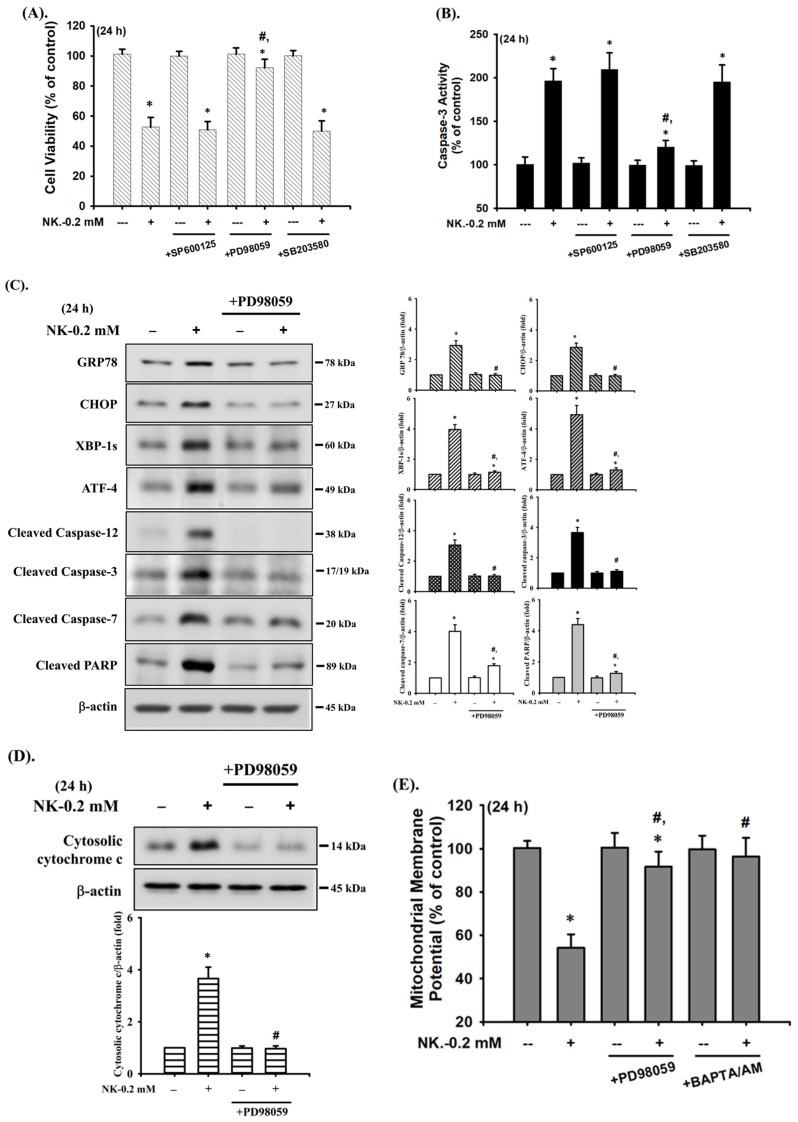
ERK1/2-mediated signaling is involved in norketamine (NK)-induced apoptosis and ER stress in RT4 cells. Cells were treated with NK (0.2 mM) for 24 h in the absence or presence of the pharmacological inhibitor of JNK (SP600125; 20 μM), ERK1/2 (PD98059; 20 μM), or p38 (SB203580; 20 μM), and then (**A**) cell viability was determined using MTT assay, and (**B**) caspase-3 activity was detected using the caspase-3 Activity Assay Kit. Furthermore, RT4 cells were treated with NK (0.2 mM) for 1 or 24 h in the absence or presence of the specific inhibitor of ERK1/2 (PD98059; 20 μM; for 1 h prior to treated with NK). The levels of protein expression for GRP78, CHOP, XBP-1s, ATF-4, caspase-3, -7, and -12, and PARP (**C**), the release of cytochrome c from the mitochondria into cytosolic fraction (**D**), and the protein phosphorylation of ERK1/2 (**F**) were examined using Western blot analysis, and (**E**) MMP depolarization was measured using flow cytometry. Data in (**A**,**B**,**E**) are presented as the means ± SD of four independent experiments assayed in triplicate. Results shown in (**C**,**D**,**F**) on representative images, and quantification was determined by densitometric analysis. Each bar presented is the mean ± SD of three independent experiments. * *p* < 0.05 compared to vehicle control. # *p* < 0.05 compared to NK treatment alone.

**Figure 8 ijms-23-04666-f008:**
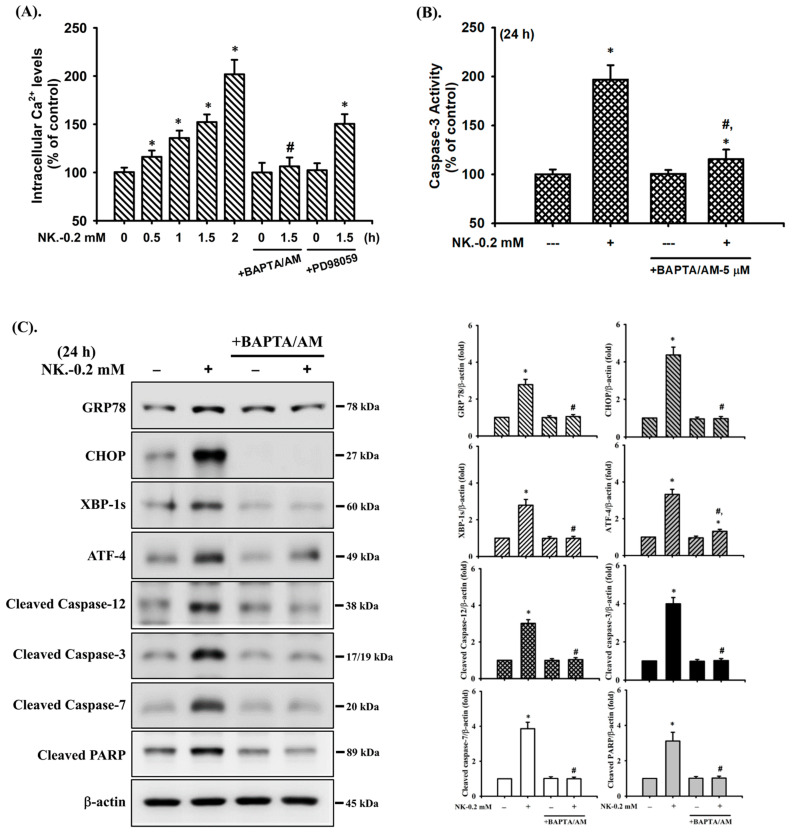
The role of [Ca^2+^]_i_ in norketamine (NK)-induced ERK1/2 signal activation and apoptosis in RT4 cells. Cells were treated with NK (0.2 mM) for different time intervals in the absence or presence of BAPTA/AM (5 μM) or PD98059 (20 μM). (**A**) [Ca^2+^]_i_ levels were determined using flow cytometry (for 0.5–1.5 h). (**B**) Caspase-3 activity was detected using the caspase-3 Activity Assay Kit (for 24 h). (**C**) The levels of protein expression for GRP78, CHOP, XBP-1s, ATF-4, caspase-3, -7, and PARP (for 24 h), (**D**) the release of cytochrome c form the mitochondria into cytosolic fraction (for 24 h), and (**E**) the protein phosphorylation for ERK1/2 (for 1 h) were examined using Western blot analysis. Data in (**A**,**B**) are presented as the means ± SD of four independent experiments assayed in triplicate. Results shown in (**C**,**D**,**E**) on representative images, and quantification was determined by densitometric analysis. Each bar presented is the mean ± SD of three independent experiments. * *p* < 0.05 compared to vehicle control. # *p* < 0.05 compared to NK treatment alone.

## Data Availability

Please contact the corresponding author for reasonable data request.

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
