# Peer review of "Norketamine, the Main Metabolite of Ketamine, Induces Mitochondria-Dependent and ER Stress-Triggered Apoptotic Death in Urothelial Cells via a Ca2+-Regulated ERK1/2-Activating Pathway"

_ijms, 2022, doi:10.3390/ijms23094666_

Round 1
Reviewer 1 Report
Authors investigated the toxic effects and possible mechanisms of the ketamine main metabolite norketamine (NK) on bladder urothelial cells. The roles of MAPKs and [Ca2+]i signals and the involvements of mitochondrial dysfunction and ER stress response in the NK-induced urothelial cytotoxicity would be examined and clarified. The manuscript is well structured and well discussed. However, some points should be checked and corrected before its acceptance in this journal.
Therefore, I recommended the publications of the paper after major revision according to given my comments.
- Title should be changed.
- The abstract is not clear. Please add the aim and objective of the MS.
- The study's background should be clearly stated. Describe the introduction and review of the work.
- Please speculate on the results. The discussion must improve.
- In Conclusion, the authors should add the significance of this research, and its potential practical application.
- The MS English needs to be improved. The article's English must be carefully checked for grammatical errors.
Author Response
Reply to Reviewer’ 1 comments:
Q1. Title should be changed.
Ans.: Thank you very much for reviewer’s comments and suggestions.
Title of our manuscript has been changed to ‘‘Norketamine, the main metabolite of ketamine, induces apoptotic cell death in the urothelial cells via a Ca2+-regulated ERK1/2-activating pathway’’ .
Q2. The abstract is not clear. Please add the aim and objective of the MS.
Ans.: We have been re-edited the section of ‘‘Abstract’’. The aim and objective of this manuscript has also been added (Line 36-38), as follows:
‘‘However, the molecular mechanisms contributing to NK-induced urothelial cytotoxicity are almost unclear. Here, we aimed to investigate the toxic effects of NK and the potential mechanisms underlying NK-induced urothelial cell injury.’’
Q3. The study's background should be clearly stated. Describe the introduction and review of the work. Please speculate on the results. The discussion must improve.
Ans.: According to reviewer’s suggest, we have been reorganized, re-edited, and rewrote our manuscript. Further suggestions on our revised manuscript will be greatly appreciated.
Q4. In Conclusion, the authors should add the significance of this research, and its potential practical application.
Ans.: We have been added ‘‘the significance of this research, and its potential practical application’’ in the ‘Conclusion’ section (Line 626-631), as follows:
‘‘These findings not only provide useful evidence supporting the role of the main ketamine metabolite NK as a key substance in ketamine abuse-induced urothelial cytotoxicity but also underline that the regulation of [Ca2+]i/ERK signaling pathways may be a promising intervention with potential application for the development of targeted drugs in the treatment of NK-induced urothelial cystitis.’’
Q5. The MS English needs to be improved. The article's English must be carefully checked for grammatical errors.
Ans.: We have been carefully corrected and proofreading our revised manuscript. Moreover, the language of our manuscript has also been revised by an English Editing Services of MDPI Author Services (English Editing ID english-42629, and as below shown ‘the certification of English-editing of revised manuscript’).

Reviewer 2 Report
In this manuscript the authors investigated the ER-stress and pro-apoptotic effect of norketamin in human urinary bladder epithelial-derived RT4 cells. This study is interesting and overall well conducted despite some drawbacks that hopefully can be corrected. The introduction is well written and extensively documented. The results are well presented in text form but suffer from the lack of quantification of western-blot data (see below). The results are well discussed but some additional points could be informative and therefore added (see below). Experimental material and methods are well described. It is therefore a real pity that the insufficient quality and the lack of quantification of the western blots do not allow to accept this manuscript as is.
Authors are encouraged to provide unsaturated (and, as a supplementary data, uncroped) western blot images as well as their quantification and statistical analysis. The effect of norketamin on oxidative stress and in particular Nrf2, which is also linked to ER stress, would be a very interesting complementary study to be added either as experimental data or, at least, in the discussion.
Specific concerns:
- Please provide uncropped blot images as supplementary data. Especially for Figure 2B where the cleaved forms of PARP and caspase should be visible.
- Why choosing to show the “pro” forms of caspase and PARP in Figure 1B with ketamin treatment and the “cleaved” forms in Figure 2B and 2C with norketamin treatment?
- The b-actin signal is overexposed in all Figures and non-saturated images should be provided. In addition, please provide a quantification for all western blot experiments with statistical analysis of replicates.
- Figure 5, “PERK” should probably be “ERK” or rather ERK-1/2. In addition, the blots for GRP94, ERK, eIF-2a, and actin are fully saturated and then not acceptable. Please provide less exposed images and quantification.
- Figure 6, why the JNK-2 total level is not shown? In addition, the blots for total JNK-1, ERK-1/2, p38 and actin are fully saturated and then not acceptable. Please provide less exposed images and quantification.
- What about Nrf2 (NFE2L2) under ketamin and norketamin treatment? Additional experiments or discussion of the literature should be added.
- What about oxidative stress under ketamin and norketamin treatment? Additional experiments or discussion of the literature should be added.
- Line 208, please correct “PRAP” to “PARP”
Author Response
Reply to Reviewer’ 2 comments:
Q1. Authors are encouraged to provide unsaturated (and, as a supplementary data, uncroped) western blot images as well as their quantification and statistical analysis.
Ans.: Thank you very much for reviewer’s comments and suggestions.
We have rechecked and provided the non-saturated Western blot images in the ‘Figures’ section, and the quantification results of western blot experiments with statistical analysis have been added in the revised manuscript. Furthermore, we have also been carefully corrected and proofreading our revised manuscript. Moreover, the language of our manuscript has also been revised by an English Editing Services of MDPI Author Services (English Editing ID english-42629, and as below shown ‘the certification of English-editing of revised manuscript’).
Q2. The effect of norketamine on oxidative stress and in particular Nrf2, which is also linked to ER stress, would be a very interesting complementary study to be added either as experimental data or, at least, in the discussion.
Ans.: We have been added ‘‘the effect of ketamine and norketamine on oxidative stress, Nrf2, and linked to ER stress’’ in the last part of ‘Discussion’ section (Line 486-515), as follows:
‘‘Oxidative stress has been demonstrated to play a critical role in several pathological conditions of the urinary bladder [73]. Several recent in vitro and in vivo studies have reported that ketamine abuse can induce reactive oxygen species (ROS) production, leading to uroepithelial cell apoptosis and death [9,74-76]. Furthermore, it is well known that the nuclear factor erythroid 2-related factor 2 (Nrf2), a member of a family of basic leucine transcription factors participating in the regulation of antioxidant response element (ARE)-mediated gene expression, is part of one of the most important protective mechanisms/antioxidant defense systems against oxidative stress in mammalian cells [77]. Nrf2 activates a series of enzymes with antioxidant and detoxifying activity that play a key role in the protection of cells against various environmental stresses, such as electrophiles, reactive oxygen species, DNA damage, and apoptosis [78]. Under normal conditions, Nrf2 is present in the cytoplasm and complexes with Kelch-like ECH-associated protein 1 (Keap1). In response to oxidative stress, Nrf2 is dissociated from Keap1 (the conformational change and the release) and then translocates into the nucleus [79]. It has been showed that an increase in Nrf2 levels can protect uroepithelial cells from chemical-induced cytotoxicity and apoptosis [80,81]. Nrf2 knockout substantially increases the susceptibility to a broad range of chemical toxicity and disease conditions associated with oxidative pathology [82,83]. A study by Sun et al. [84] has shown the positive correlation between the loss of Nrf2 and the exacerbation of ER stress-induced apoptosis in the brain tissue of Nrf2 knockout mice with traumatic brain injury. Recently, Liu et al. [9] and Cui et al. [32] reported that ketamine exposure-induced uroepithelial cell apoptosis involves oxidative stress and ER stress responses. Our results from this study also found that NK, similarly to ketamine [9,32], could significantly increase ROS generation (Supplement Figure S1) and the expression of ER stress-related molecules (Figure 5). Both ketamine and norketamine treatment also led to a marked decrease in Nrf2 protein levels (Supplement Figure S2), suggesting that Nrf2 might play a key role in ketamine- and NK-induced uroepithelial cell apoptosis. However, the relationship between oxidative stress, Nrf2, and ER stress underlying the ketamine- and NK-induced uroepithelial cell injury are mostly unclear. Thus, further experiments are required to investigate this important issue in the future.’’
Q3. Please provide uncropped blot images as supplementary data. Especially for Figure 2B where the cleaved forms of PARP and caspase should be visible.
Ans.: We have rechecked and provided the “cleaved forms of caspase and PARP” in Figure 2B with ketamine treatment. Furthermore, the quantification results of western blot experiments with statistical analysis have also been added.
Q4. Why choosing to show the “pro” forms of caspase and PARP in Figure 2B with ketamine treatment and the “cleaved” forms in Figure 3B and 3C with norketamin treatment?
Ans.: We have rechecked and provided the “cleaved forms of caspase-3/-7/-9 and PARP” in Figure 2B with ketamine treatment. Furthermore, the quantification results of western blot experiments with statistical analysis have also been added.
Q5. The b-actin signal is overexposed in all Figures and non-saturated images should be provided. In addition, please provide a quantification for all western blot experiments with statistical analysis of replicates.
Ans.: We have been rechecked and provided the non-saturated Western blot images in the ‘Figures’ section. Furthermore, the quantification results of western blot experiments with statistical analysis have also been added.
Q6. Figure 5, “PERK” should probably be “ERK” or rather ERK-1/2. In addition, the blots for GRP94, ERK, eIF-2a, and actin are fully saturated and then not acceptable. Please provide less exposed images and quantification.
Ans.:
- “PERK” is “protein kinase RNA-like endoplasmic reticulum kinase”, which is one of ER stress-related proteins.
- We have been rechecked and provided the less exposed blot images for GRP94, PERK, eIF-2a, and b-actin; and the quantification results of western blot experiments with statistical analysis have also been added.
Q7. Figure 6, why the JNK-2 total level is not shown? In addition, the blots for total JNK-1, ERK-1/2, p38 and actin are fully saturated and then not acceptable. Please provide less exposed images and quantification.
Ans.:
- Blot image of the JNK-2 total is been added in Figure 6.
- We have also been rechecked and provided the non-saturated blot images for total JNK-1, ERK-1/2, p38, and b-actin; and the quantification results of western blot experiments with statistical analysis have also been added.
Q8-1. What about Nrf2 (NFE2L2) under ketamine and norketamine treatment? Additional experiments or discussion of the literature should be added.
Q8-2.What about oxidative stress under ketamine and norketamine treatment? Additional experiments or discussion of the literature should be added.
Ans.: We have been added ‘‘the effect of ketamine and norketamine on oxidative stress, Nrf2, and linked to ER stress’’ in the last part of ‘Discussion’ section (Line 486-515), as follows:
‘‘Oxidative stress has been demonstrated to play a critical role in several pathological conditions of the urinary bladder [73]. Several recent in vitro and in vivo studies have reported that ketamine abuse can induce reactive oxygen species (ROS) production, leading to uroepithelial cell apoptosis and death [9,74-76]. Furthermore, it is well known that the nuclear factor erythroid 2-related factor 2 (Nrf2), a member of a family of basic leucine transcription factors participating in the regulation of antioxidant response element (ARE)-mediated gene expression, is part of one of the most important protective mechanisms/antioxidant defense systems against oxidative stress in mammalian cells [77]. Nrf2 activates a series of enzymes with antioxidant and detoxifying activity that play a key role in the protection of cells against various environmental stresses, such as electrophiles, reactive oxygen species, DNA damage, and apoptosis [78]. Under normal conditions, Nrf2 is present in the cytoplasm and complexes with Kelch-like ECH-associated protein 1 (Keap1). In response to oxidative stress, Nrf2 is dissociated from Keap1 (the conformational change and the release) and then translocates into the nucleus [79]. It has been showed that an increase in Nrf2 levels can protect uroepithelial cells from chemical-induced cytotoxicity and apoptosis [80,81]. Nrf2 knockout substantially increases the susceptibility to a broad range of chemical toxicity and disease conditions associated with oxidative pathology [82,83]. A study by Sun et al. [84] has shown the positive correlation between the loss of Nrf2 and the exacerbation of ER stress-induced apoptosis in the brain tissue of Nrf2 knockout mice with traumatic brain injury. Recently, Liu et al. [9] and Cui et al. [32] reported that ketamine exposure-induced uroepithelial cell apoptosis involves oxidative stress and ER stress responses. Our results from this study also found that NK, similarly to ketamine [9,32], could significantly increase ROS generation (Supplement Figure S1) and the expression of ER stress-related molecules (Figure 5). Both ketamine and norketamine treatment also led to a marked decrease in Nrf2 protein levels (Supplement Figure S2), suggesting that Nrf2 might play a key role in ketamine- and NK-induced uroepithelial cell apoptosis. However, the relationship between oxidative stress, Nrf2, and ER stress underlying the ketamine- and NK-induced uroepithelial cell injury are mostly unclear. Thus, further experiments are required to investigate this important issue in the future.’’
Q9. Line 208, please correct “PRAP” to “PARP”
Ans.: This mistake has been corrected.

Round 2
Reviewer 1 Report
Requested corrections were completed.
Author Response
Reply to Reviewer’ 1 comments:
Q1. Requested corrections were completed.
Ans.: We thanked very much for reviewer’s affirmation.
Reviewer 2 Report
This revised manuscript addressed most of the reviewer’s recommendations, especially western-blot quantification and discussion about oxidative stress and Nrf2. I would like to thank the authors for these important corrections and some other minor points.
However, the authors did not provide the uncropped/uncut WB images (the photographs available in supplement are still truncated versions of the blots) and most of the images provided are very heavily overexposed (especially for beta-actin, but not only) which makes the quantifications quite improbable and unconvincing.
This problem of overexposed blots is for beta-actin in Figures 2B, 3B, 3C, 4C, 5A, 5B, 5C, 6, 7C, 7D, 7F, 8C, 8D, 8E; and also for GRP94 (Fig. 5A), PERK and eIF-2alpha (Fig. 5B), JNK2, JNK1, ERK1/2 and p38 MAPK in Figure 6, and ERK1/2 in Figure 7F.
So, the authors are requested again to provide less exposed images of all saturated blot (see the list above) for the figures to be included in the main text and uncropped/uncut blots as supplementary data (especially for caspase and PARP where both the cleaved and uncleaved forms should be visible).
Author Response
Reply to Reviewer’ 2 comments:
Q1. This revised manuscript addressed most of the reviewer’s recommendations, especially western-blot quantification and discussion about oxidative stress and Nrf2. I would like to thank the authors for these important corrections and some other minor points. However, the authors did not provide the uncropped/uncut WB images (the photographs available in supplement are still truncated versions of the blots) and most of the images provided are very heavily overexposed (especially for beta-actin, but not only) which makes the quantifications quite improbable and unconvincing. This problem of overexposed blots is for beta-actin in Figures 2B, 3B, 3C, 4C, 5A, 5B, 5C, 6, 7C, 7D, 7F, 8C, 8D, 8E; and also for GRP94 (Fig. 5A), PERK and eIF-2alpha (Fig. 5B), JNK2, JNK1, ERK1/2 and p38 MAPK in Figure 6, and ERK1/2 in Figure 7F. So, the authors are requested again to provide less exposed images of all saturated blot (see the list above) for the figures to be included in the main text and uncropped/uncut blots as supplementary data (especially for caspase and PARP where both the cleaved and uncleaved forms should be visible).
Ans.: Thank you very much for reviewer’s comments and suggestions.
- We have been rechecked and provided the less exposed Western blot images (including b-actin in Figures 2B, 3B, 3C, 4C, 5A, 5B, 5C, 6, 7C, 7D, 7F, 8C, 8D, 8E; and also for GRP94 (Fig. 5A), PERK and eIF-2alpha (Fig. 5B), JNK2, JNK1, ERK1/2 and p38 MAPK in Figure 6, and ERK1/2 in Figure 7F) in the ‘Figures’ section. Furthermore, the quantification results of western blot experiments with statistical analysis have also been rechecked and corrected.
- The uncropped blot images as supplementary data have been provided.
- In WB images of caspase-3/-7/-9 and PARP, we focused on the expression of the cleaved form. Thus, we used the antibodies specific for cleaved caspase-3 (Cat. No.: #9661), cleaved caspase-7 (Cat. No.: #9491), cleaved caspase-9 (Cat. No.: #7237), which were purchased from Cell Signaling Technology (Cell Signaling Technology, Danvers, MA, USA). The images of caapase-3/-7/-9 were showed the cleaved form, without the uncleaved form.

Round 3
Reviewer 2 Report
OK